# PhyDOSE: Design of follow-up single-cell sequencing experiments of tumors

**Leah L. Weber**[1⊙], **Nuraini Aguse**[1⊙], **Nicholas Chia**[2,3], **Mohammed El-Kebir**[1] *

**1** Dept. of Computer Science, University of Illinois at Urbana-Champaign, Urbana, Illinois, United States of America, **2** Microbiome Program, Center for Individualized Medicine, Mayo Clinic, Rochester, Minnesota, United States of America, **3** Division of Surgical Research, Department of Surgery, Mayo Clinic, Rochester, Minnesota, United States of America

⊙ These authors contributed equally to this work.
* melkebir@illinois.edu

**Data Availability Statement:** Simulated and real data are available at https://github.com/elkebir-group/PhyDOSE. Source code and R package are available at https://github.com/elkebir-group/phydoser (under AGPL-3.0 license).

## Abstract

The combination of bulk and single-cell DNA sequencing data of the same tumor enables the inference of high-fidelity phylogenies that form the input to many important downstream analyses in cancer genomics. While many studies simultaneously perform bulk and single-cell sequencing, some studies have analyzed initial bulk data to identify which mutations to target in a follow-up single-cell sequencing experiment, thereby decreasing cost. Bulk data provide an additional untapped source of valuable information, composed of candidate phylogenies and associated clonal prevalence. Here, we introduce PhyDOSE, a method that uses this information to strategically optimize the design of follow-up single cell experiments. Underpinning our method is the observation that only a small number of clones uniquely distinguish one candidate tree from all other trees. We incorporate distinguishing features into a probabilistic model that infers the number of cells to sequence so as to confidently reconstruct the phylogeny of the tumor. We validate PhyDOSE using simulations and a retrospective analysis of a leukemia patient, concluding that PhyDOSE's computed number of cells resolves tree ambiguity even in the presence of typical single-cell sequencing errors. We also conduct a retrospective analysis on an acute myeloid leukemia cohort, demonstrating the potential to achieve similar results with a significant reduction in the number of cells sequenced. In a prospective analysis, we demonstrate the advantage of selecting cells to sequence across multiple biopsies and that only a small number of cells suffice to disambiguate the solution space of trees in a recent lung cancer cohort. In summary, PhyDOSE proposes cost-efficient single-cell sequencing experiments that yield high-fidelity phylogenies, which will improve downstream analyses aimed at deepening our understanding of cancer biology.

## Author summary

Cancer development in a patient can be explained using a phylogeny—a tree that describes the evolutionary history of a tumor and has therapeutic implications. A tumor phylogeny is constructed from sequencing data, commonly obtained using either bulk or

**Funding:** L.L.W., N.A., N.C. and M.E.K. were supported by UIUC Center for Computational Biotechnology and Genomic Medicine (grant: CSN 1624790). M.E.K. was supported by the National Science Foundation (grant: CCF 1850502). The funders had no role in study design, data collection and analysis, decision to publish, or preparation of the manuscript.

**Competing interests:** The authors have declared that no competing interests exist.

single-cell DNA sequencing technology. The accuracy of tumor phylogeny inference increases when both types of data are used, but single-cell sequencing may become prohibitively costly with increasing number of cells. Here, we propose a method that uses bulk sequencing data to guide the design of a follow-up single-cell sequencing experiment. Our results suggest that PhyDOSE provides a significant decrease in the number of cells to sequence compared to the number of cells sequenced in existing studies. The ability to make informed decisions based on prior data can help reduce the cost of follow-up single cell sequencing experiments of tumors, improving accuracy of tumor phylogeny inference and ultimately getting us closer to understanding and treating cancer.

This is a *PLOS Computational Biology* Methods paper.

## Introduction

Tumorigenesis follows an evolutionary process during which cells gain and accumulate somatic mutations that lead to cancer [1]. The most natural expression of an evolutionary process is a *phylogeny*—a tree that describes the order and branching points of events in the history of a cellular population. Tumor phylogenies are critical to understanding and ultimately treating cancer, with recent studies using tumor phylogenies to identify mutations that drive cancer progression [2, 3], assess the interplay between the immune system and the clonal architecture of a tumor [4, 5], and identify common evolutionary patterns in tumorigenesis and metastasis [6, 7]. These downstream analyses critically rely on accurate phylogenies that are inferred from sequencing data of a tumor.

The majority of current cancer genomics data consist of pairs of matched normal and tumor samples that have undergone bulk DNA sequencing. Bulk data is composed of sequences from cells with distinct genomes. More specifically, we observe frequencies $\mathbf{f} = [f_i]$ for the set of somatic mutations in the tumor (Fig 1A). Many deconvolution methods have been proposed for tumor phylogeny inference from such data [8–13], typically inferring a set $\mathcal{T}$ of equally plausible trees (Fig 1B). These approaches are unsatisfactory, as candidate trees with different topologies may alter conclusions in downstream analyses. Single-cell sequencing (SCS), as opposed to bulk sequencing, enables us to observe specific clones present within the tumor. These clones correspond to the leaves of the true phylogeny, allowing phylogeny inference methods to reconstruct the tree itself once we observe all clones in the tumor [14–17]. However, the elevated error rates of SCS, as well as its high cost [18], make it prohibitive as a standalone method for phylogeny inference. As such, hybrid methods have been recently proposed to infer high-fidelity phylogenies from combined bulk and SCS data obtained from the same tumor [19, 20]. Furthermore, without a rigorous framework to determine the number of single-cells to sequence, this decision is currently guided by budget constraints or arbitrarily determined by exogenous factors, such as thresholds for a sequencing run. This could result in excessive costs by sequencing too many cells or sunk costs associated with an unsuccessful experiment when an insufficient number of cells are sequenced.

Several hybrid datasets have been obtained by performing bulk and single-cell DNA sequencing simultaneously [21, 22]. However, there is merit in first performing bulk sequencing to guide follow-up SCS experiments. For instance, several studies first identified a subset of single-nucleotide variants from the bulk data to target in subsequent SCS experiments, thereby reducing costs compared to conventional whole-genome SCS approaches [23–25]. A recently

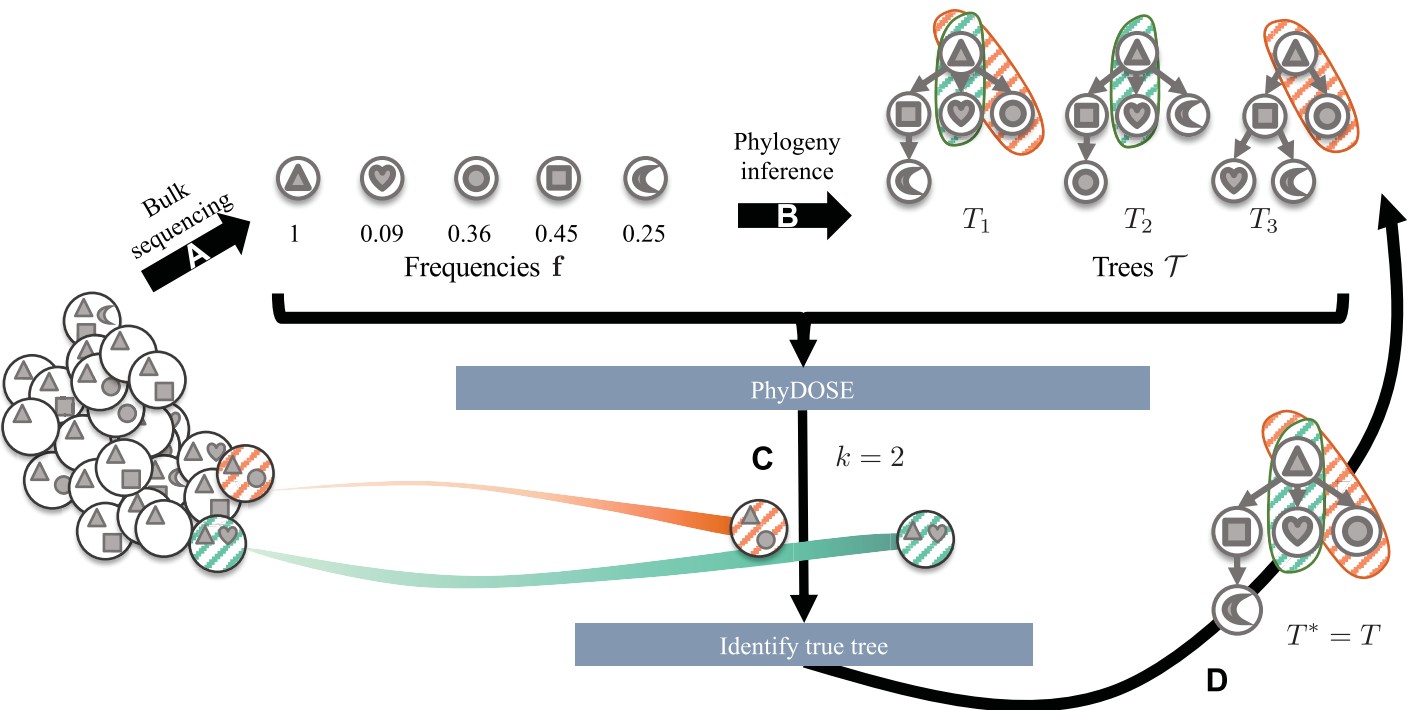

**Fig 1. PhyDOSE computes the number of single cells to sequence to identify the true phylogeny.** (A) Mutation frequencies **f** obtained from bulk DNA sequencing data. (B) The solution space $\mathcal{T}$ of trees inferred from **f**. We show a distinguishing feature of $T_1$ (orange and green). (C) For tree $T_1$, PhyDOSE suggests that $k = 2$ single cells suffice to observe clones that are unique to $T_1$. (D) In a follow-up SCS experiment we observe $k = 2$ cells, one from the orange clone and one from the green clone. As such, we eliminate trees $T_2$ and $T_3$, concluding that phylogeny $T_1$ is the true phylogeny $T^*$.

introduced method, SCOPIT, computes how many cells are needed to observe all clones of a tumor, given estimates on the smallest prevalence of a clone as well as the number of clones with that smallest prevalence to detect [26]. The authors provide no guidance on how to obtain these two quantities. Here, we build upon this work by directly incorporating knowledge encoded by the trees $\mathcal{T}$ inferred from the initial bulk sequencing data. Indeed, by using data from an SCS experiment we may eliminate trees from $\mathcal{T}$ that do not align with the observed clones (Fig 1). In other words, if we observe all clones in a tumor, it is possible to determine the phylogeny of the tumor. However, is it possible to achieve the same goal by observing fewer clones? If so, how many cells are necessary for us to observe the required clones?

We introduce <u>Phy</u>logenetic <u>D</u>esign <u>O</u>f <u>S</u>ingle-cell sequencing <u>E</u>xperiments (PhyDOSE), a method to strategically design a follow-up SCS experiment aimed at inferring the true phylogeny (Fig 1). Given a set $\mathcal{T}$ of candidate trees inferred from initial bulk data, we describe how to distinguish a single tree $T$ among the rest using features unique to $T$. In particular, if our SCS experiment results in observing cells corresponding to a distinguishing feature of $T$, we may conclude that $T$ is in fact the true tree. This means that we can typically identify $T$ using only a subset of the clones. To determine the number of cells to sequence, we introduce a probabilistic model that incorporates SCS errors and models successful SCS experiments as a tail probability of a multinomial distribution (Fig 1D). Finally, we reconcile the sampled cells utilizing these distinguishing features to infer the true phylogeny (Fig 1D) and provide heuristics for considering uncertainty in frequency estimates and determining the number of cells to sequence across multiple available biopsies. We validate PhyDOSE using both simulated data and a retrospective analysis of a leukemia patient that has undergone both bulk and SCS sequencing. We also demonstrate the utility of PhyDOSE by prospectively computing how

many cells are needed to resolve the uncertainty in phylogenies of a recent acute myeloid leukemia cohort [27] and lung cancer cohort [3]. The cost-efficient SCS experiments enabled by PhyDOSE will yield high-fidelity phylogenies, improving downstream analyses aimed at understanding tumorigenesis and developing treatment plans.

## Materials and methods

We introduce <u>Phy</u>logenetic <u>D</u>esign <u>O</u>f <u>S</u>ingle-cell sequencing <u>E</u>xperiments (PhyDOSE), a method to determine the number of single cells to sequence to identify the true phylogeny given initial bulk sequencing data. PhyDOSE is implemented in C++/R and is available as an R package at https://github.com/elkebir-group/phydoser. This section describes the various methodological components of PhyDOSE.

### Problem statement

Let $n$ be the number of single-nucleotide variants, or simply *mutations*, identified from initial bulk sequencing data of a matched normal and tumor biopsy sample. For each mutation $i$, we observe the *variant allele frequency* (VAF), i.e. the fraction of aligned reads that harbor the tumor allele at the locus of mutation $i$. Specialized methods exist that combine copy number information and VAFs to infer a *cancer cell fraction* $f_i$ for each mutation $i$, which is the proportion of cells in the tumor biopsy that contain at least one copy of the mutation [3, 28–30]. Here, we refer to cancer cell fractions as *frequencies*. Typically, phylogenies $\mathcal{T}$ inferred by current methods from frequencies $\mathbf{f} = [f_i]$ adhere to the infinite sites assumption. That is, each mutation $i$ is introduced exactly once at vertex $v_i$ and never subsequently lost.

When we sequence a single cell from the same tumor biopsy, assuming no errors, we identify a clone of the tumor. In other words, we observe a set of mutations that must form a connected path in the unknown true phylogeny $T^*$. By repeatedly sequencing single cells until we observe all clones in the tumor, we will have observed all root-to-vertex paths of $T^*$, thus identifying tree $T^*$ itself. We assume that (i) the true unknown phylogeny $T^*$ is among the trees in $\mathcal{T}$ and that (ii) mutations among single cells that we sample from the tumor biopsy follow the same distribution as $\mathbf{f}$. These assumptions are important for the mathematical derivation of PhyDOSE but it is typical for violations to occur in practice. Through simulations, we explore the impact of violating these assumptions and show that our approach is robust to many realistic scenarios.

This leads to the following question and problem statement with respect to these two assumptions. How many single cells do we need to identify $T^*$ with confidence level $\gamma$?

**Problem 1** (SCS POWER CALCULATION (SCS-PC)). Given a set $\mathcal{T}$ of candidate phylogenies, frequencies $\mathbf{f}$ and confidence level $\gamma$, find the minimum number $k^*$ of single cells needed to determine the true phylogeny $T^*$ among $\mathcal{T}$ with probability at least $\gamma$.

Clearly, we do not know which phylogeny in $\mathcal{T}$ is the true underlying phylogeny $T^*$ of the tumor. Thus, we consider a slightly different problem: In the $T$-SCS-PC problem (defined formally at the end of the section), we are given an arbitrary phylogeny $T \in \mathcal{T}$ and want to perform a similar power calculation when conditioning on $T$ being the true phylogeny. By solving the $T$-SCS-PC problem for all trees $T_1, \ldots, T_{|\mathcal{T}|}$, we obtain the numbers $k(T_1), \ldots, k(T_{|\mathcal{T}|})$ of single cells needed for each tree. As $T^*$ is in $\mathcal{T}$, the maximum number among $k(T_1), \ldots, k(T_{|\mathcal{T}|})$ is an upper bound on the number of required SCS experiments to identify $T^*$ with probability at least $\gamma$. To solve the $T$-SCS-PC problem, we need to reason for which SCS experiments we can conclude that $T$ is the true phylogeny.

Observe that each tree $T$ in $\mathcal{T}$ describes a unique set of clones, corresponding to the sets of mutations encountered in all root-to-vertex paths of $T$ (Fig 1). Thus, if we observe all clones of

a phylogeny $T$ in our SCS experiments, we may conclude that $T$ is the true phylogeny. What is the probability of doing so? To answer this question, we must compute the prevalence of each clone in the tumor biopsy.

For phylogenies that adhere to the infinite sites assumption, the *prevalence* $\mathbf{u}(T, \mathbf{f}) = [u_i]$ of the clones in the tumor biopsy are uniquely determined by the phylogeny $T$ and frequencies $\mathbf{f}$ as

$$u_i = f_i - \sum_{j \in \delta_T(i)} f_j \qquad \forall i \in [n]. \tag{1}$$

where $\delta_T(i)$ is the set of children of the node where mutation $i$ was introduced [9].

Tumor phylogeny inference methods guarantee that the inferred phylogenies $\mathcal{T}$ from frequencies $\mathbf{f}$ have clonal prevalence $\mathbf{u}(T, \mathbf{f}) = [u_i]$ that are nonnegative and that $\sum_{i=1}^{n} u_i \leq 1$, where the remainder $u_0 = 1 - \sum_{i=1}^{n} u_i$ is the prevalence of the normal clone. Thus, conditioning on a phylogeny $T$ and frequencies $\mathbf{f}$, sequencing one cell from the tumor will lead us to observe one of the $n + 1$ clones of $T$ with probabilities $(u_0, \ldots, u_n)$. In other words, the outcome of this SCS experiment with one cell is a draw from the categorical distribution $\mathrm{Cat}(u_0, \ldots, u_n)$. The possible outcomes of an SCS experiment composed of $k$ cells thus follow a multinomial distribution $\mathrm{Mult}(u_0, \ldots, u_n)$. Thus, the probability of observing all tumor clones of $T$ in such an SCS experiment with $k$ cells corresponds to the tail probability of the multinomial where each of the $n$ tumor clones is observed at least once.

The corresponding *power calculation* is to determine the smallest number for $k$ where the tail probability is greater or equal to the confidence level $\gamma$. Note that this power calculation for observing all clones has been previously introduced [26].

Importantly, in many cases we need not observe all clones of $T$ to distinguish $T$ from the remaining phylogenies $\mathcal{T} \setminus \{T\}$ (Fig 2). This means that we may conclude that $T$ is the true phylogeny with an SCS experiment with fewer cells. To formalize this notion, we start by defining a featurette.

**Definition 1**. A *featurette* $\tau$ is a subset of mutations.

We say that a featurette $\tau$ is *present* in a phylogeny $T$ if the nodes/mutations of $\tau$ form a connected path of $T$ starting at the root node, otherwise we say that $\tau$ is *absent* in $T$. The same featurette, however, may be present in more that one phylogeny. Thus, multiple featurettes may be required to distinguish a phylogeny $T$ from the remaining phylogenies $\mathcal{T} \setminus \{T\}$.

**Definition 2**. A set $\Pi$ of featurettes is a *distinguishing feature of $T$* if (i) for all featurettes $\tau \in \Pi$ it holds that $\tau$ is present in $T$, and (ii) for each remaining phylogeny $T' \in \mathcal{T} \setminus \{T\}$ there exists a featurette $\tau' \in \Pi$ where $\tau'$ is absent in $T'$.

Thus, an SCS experiment where we observe one cell from each clone of a distinguishing feature $\Pi$ of $T$ enables us to conclude that phylogeny $T$ is the true phylogeny. As discussed, every phylogeny $T$ has a *trivial distinguishing feature*, which is composed of all featurettes present in $T$. Moreover, $T$ may have multiple distinguishing features. Therefore, we must consider the complete set of all distinguishing features, which we call the distinguishing feature family.

**Definition 3**. The set $\Phi(T, \mathcal{T} \setminus \{T\})$ composed of all distinguishing features of $T$ with respect to $\mathcal{T} \setminus \{T\}$ is a *distinguishing feature family of $T$*.

Let $(c_0, \ldots, c_n)$ be the outcome of an SCS experiment of $k$ cells, where $c_i \geq 0$ is the number of cells observed of clone $i$ and $\sum_{i=0}^{n} c_i = k$. This experiment is *successful* if, among the $k$ sequenced cells, we observe the clones of at least one distinguishing feature $\Pi \in \Phi(T, \mathcal{T} \setminus \{T\})$—i.e. $c_i > 0$ for all clones $i$ in some distinguishing feature $\Pi \in \Phi(T, \mathcal{T} \setminus \{T\})$. As discussed, conditioning on frequencies $\mathbf{f}$ and $T$ being the true phylogeny, outcomes $(c_0, \ldots, c_n)$ of SCS experiments of $k$ cells follow a multinomial distribution Mult

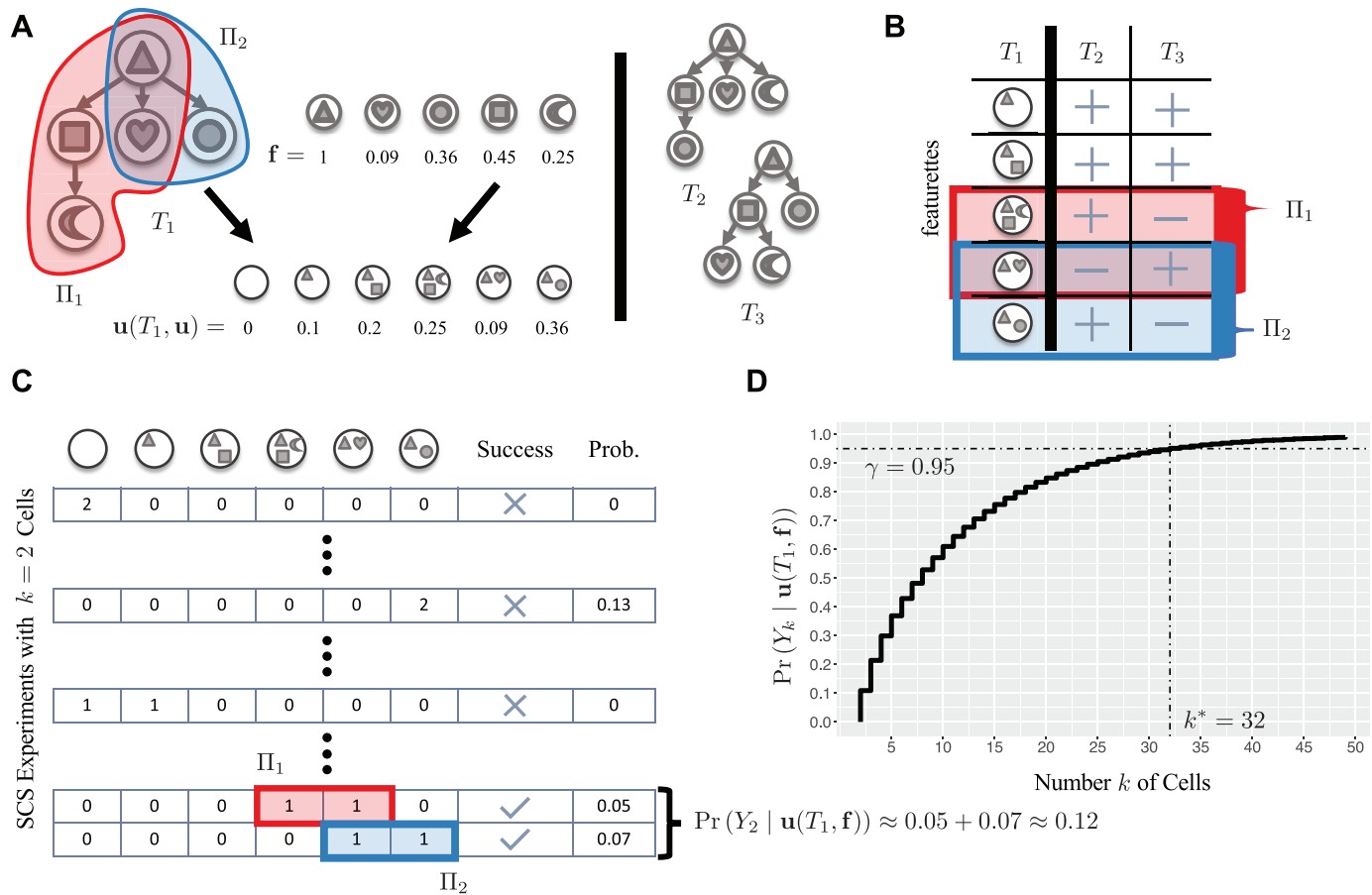

**Fig 2. The SCS power calculation for phylogeny $T$ ($T$-SCS-PC) problem.** (A) We are given frequencies $\mathbf{f}$ and a tree $T_1$ that we want to distinguish from the other trees $\{T_2, T_3\}$. The pair $(T_1, \mathbf{f})$ uniquely determine clonal prevalence $\mathbf{u}(T_1, \mathbf{f})$. (B) Featurettes of $T_1$ correspond to root-to-vertex paths, yielding distinguishing features $\Pi_1$ and $\Pi_2$, each with one featurette absent in $T_2$ and another absent in $T_3$. (C) With $k = 2$ cells, we must observe clones from either $\Pi_1$ or $\Pi_2$ for a successful outcome, resulting in probability $\Pr(Y_2|\mathbf{u}(T_1, \mathbf{f})) \approx 0.12$. (d) To increase this probability to $\gamma = 0.95$, we need $k^* = 32$ cells.

$(k, u_0, \ldots, u_n)$ where $\mathbf{u}(T, \mathbf{f}) = [u_i]$ is defined as in (1). Let $Y_k$ denote the event of a successful outcome. We are interested in computing the probability $\Pr(Y_k|\mathbf{u}(T, \mathbf{f}))$, which equals the sum of the probabilities of all successful outcomes. More specifically, we want to determine the smallest number $k^*$ of single cells to sequence such that $\Pr(Y_{k^*}|\mathbf{u}(T, \mathbf{f}))$ is at least the prescribed confidence level $\gamma$ (Fig 2).

**Problem 2** (SCS POWER CALCULATION FOR PHYLOGENY $T$ ($T$-SCS-PC)). Given a set $\mathcal{T}$ of candidate phylogenies and a phylogeny $T \in \mathcal{T}$, frequencies $\mathbf{f}$ and confidence level $\gamma$, find the minimum number $k^*$ of single cells needed such that $\Pr(Y_{k^*}|\mathbf{u}(T, \mathbf{f})) \geq \gamma$.

In Section A.1 in S1 Text, we prove that the above problem is NP-hard.

**Theorem 1**. $T$-SCS-PC is NP-hard.

**Multiple biopsies.** The SCS-PC problem is only applicable to bulk sequencing data obtained from a single biopsy, i.e. the number of cells calculated is only for an SCS experiment on one sample. However, bulk samples from tumors are often obtained from multiple biopsies, each with different mutation frequencies and consequently different clonal prevalences. One approach to support such data is to solve the SCS-PC problem for each biopsy in isolation and select the biopsy that requires the smallest number of cells. However, a more cost-effective approach that also better captures intra-tumor heterogeneity is to perform a follow-up SCS

experiment with cells from multiple biopsies. In particular, the naive selection approach might not yield a solution if the clones of a distinguishing feature do not co-occur in a single biopsy.

With $b \geq 1$ biopsies, the input changes from a frequency vector $\mathbf{f} \in [0, 1]^n$ to a frequency matrix $F \in [0, 1]^{b \times n}$, whose entries $f_{pi}$ indicate the frequency of mutation $i$ in biopsy $p$. Similarly, the output changes from an integer $k^* \in \mathbb{N}$ to a count vector $\mathbf{k}^* \in \mathbb{N}^b$, such that each entry $k_p^*$ indicates the number of single cells in biopsy $p$. We have the following two generalizations of the SCS-PC problem and the $T$-SCS-PC problem for the case of $b \geq 1$ biopsies.

**Problem 3** (MULTI-SAMPLE SCS POWER CALCULATION (MUL-SCS-PC)). Given a set $\mathcal{T}$ of candidate phylogenies, frequencies $F$ from $b$ biopsies and confidence level $\gamma$, find the numbers $\mathbf{k}^* = [k_1^*, \ldots, k_b^*]$ of single cells needed from each biopsy to determine the true phylogeny $T^*$ among $\mathcal{T}$ with probability at least $\gamma$ and the total number $\|\mathbf{k}^*\|_1 = \sum_{p=1}^b k_p^*$ of cells is minimum.

**Problem 4** (MULTI-SAMPLE SCS POWER CALCULATION FOR PHYLOGENY $T$ ($T$-MUL-SCS-PC)). Given a set $\mathcal{T}$ of candidate phylogenies and a phylogeny $T \in \mathcal{T}$, frequencies $F$ from $b$ biopsies and confidence level $\gamma$, find the numbers $\mathbf{k}^* = [k_1^*, \ldots, k_b^*]$ of single cells needed from each biopsy such that $\Pr(Y_{\mathbf{k}} | U(T, F)) \geq \gamma$ and the total number $\|\mathbf{k}^*\|_1 = \sum_{p=1}^b k_p^*$ of cells is minimum.

For the case where $b = 1$, the $T$-SCS-PC and the $T$-MUL-SCS-PC problems are identical, amounting to following hardness result.

**Corollary 1**. $T$-MUL-SCS-PC is NP-hard.

## Multinomial power calculation

To solve the $T$-SCS-PC problem, it suffices to have an algorithm that computes $\Pr(Y_k | \mathbf{u}(T, \mathbf{f}))$, which is the probability of concluding that $T$ is the true phylogeny. Using this algorithm we identify $k^*$ by starting from $k = 0$ and simply incrementing $k$ until the corresponding probability $\Pr(Y_k | \mathbf{u}(T, \mathbf{f}))$ exceeds the prescribed confidence level $\gamma$. In the following, we describe how to efficiently compute $\Pr(Y_k | \mathbf{u}(T, \mathbf{f}))$.

Recall that the outcome of an SCS experiment composed of $k$ cells corresponds to a vector $\mathbf{c} = [c_i]$, where $c_i \geq 0$ is the number of cells that we observe from clone $i$ and $\sum_{i=0}^n c_i = k$. In a successful outcome $\mathbf{c}$ we observe at least one cell for each featurette in at least one distinguishing feature $\Pi \in \Phi(T, \mathcal{T} \setminus \{T\})$, where $\Phi(T, \mathcal{T} \setminus \{T\})$ is the distinguishing feature family. For brevity, we will write $\Phi$ rather than $\Phi(T, \mathcal{T} \setminus \{T\})$.

Let $\mathbf{c}(\Pi, k)$ denote the set of all outcomes where we observe at least one cell for each featurette in a distinguishing feature $\Pi$—i.e. $\sum_{i=0}^n c_i = k$, and for all $i \in \{0, \ldots, n\}$ it holds that $c_i > 0$ if clone $i$ is a featurette in $\Pi$ and $c_i \geq 0$ otherwise. The set $\mathbf{c}(\Phi, k)$ of successful outcomes is defined as the union $\bigcup_{\Pi \in \Phi} \mathbf{c}(\Pi, k)$. The probability of any SCS outcome $\mathbf{c} = (c_0, \ldots, c_n)$ is distributed according to $\mathrm{Mult}(k, \mathbf{u}(T, \mathbf{f}))$. Since successful outcomes enable us to conclude that $T$ is the true phylogeny, we have

$$\Pr\left(Y_k \mid \mathbf{u}(T, \mathbf{f})\right) = \sum_{\ell \in \mathbf{c}(\Phi, k)} \mathrm{Mult}(\ell \mid k, \mathbf{u}(T, \mathbf{f})) = \sum_{\ell \in \mathbf{c}(\Phi, k)} \frac{k!}{\prod_{i=0}^n \ell_i!} \prod_{i=0}^n u_i^{\ell_i}. \quad (2)$$

If there is only one distinguishing feature $\Pi$, i.e. $\Phi = \{\Pi\}$, then the desired probability is a standard tail probability of the multinomial where we sum up the probabilities of outcomes $\mathbf{c}(\Pi, k) = [c_i]$ such that $\sum_{i=0}^n c_i = k$, $c_i > 0$ if clone $i$ is a featurette of $\Pi$ and $c_i \geq 0$ otherwise. A fast calculation of this tail probability was developed using a connection to the conditional probability of independent Poisson random variables [26, 31]. If there are multiple distinguishing features but they are pairwise disjoint—i.e. no two distinct distinguishing features

share the same featurette—then we simply have

$$\Pr\left(Y_k \mid \mathbf{u}(T, \mathbf{f})\right) = \sum_{\Pi \in \Phi} \sum_{\ell \in \mathbf{c}(\Pi, k)} \mathrm{Mult}(\ell \mid k, \mathbf{u}(T, \mathbf{f})),\tag{3}$$

and we can apply the fast computation [26] to obtain each independent tail probability. However, the equality in the above equation does not hold if the family $\Phi$ is composed of distinguishing features with overlapping featurettes. Incorrectly applying this equation will lead us to overestimate the value of $k^*$. Since single-cell sequencing is expensive, overestimating the number of cells to sequence in an SCS experiment can be costly and unnecessary. One naive way would be to simply brute force all $(n + 1)^k$ SCS outcomes, but this will not scale. Instead, to calculate $\Pr(Y_k|\mathbf{u}(T, \mathbf{f}))$ exactly, we propose to use the inclusion-exclusion principle as follows.

$$\Pr\left(Y_k \mid \mathbf{u}(T, \mathbf{f})\right) = \sum_{\emptyset \subsetneq \Phi' \subsetneq \Phi} (-1)^{|\Phi'|+1} \sum_{\ell \in \mathbf{c}(I(\Phi'), k)} \mathrm{Mult}(\ell \mid k, \mathbf{u}(T, \mathbf{f})),\tag{4}$$

where $I(\Phi')$ is the set of all featurettes in $\Phi'$, i.e. $I(\Phi') = \bigcup_{\Pi \in \Phi'} \Pi$ (Fig 3A).

Thus, we need to compute $2^{|\Phi|} - 1$ tail probabilities, which each can be done using the fast calculation in SCOPIT [26].

In the worst case, $\Phi$ has $O(2^n)$ distinguishing features resulting in $O(2^n)$ tail probabilities. We now describe one final optimization that will significantly reduce the number of required computations. This is based on the following observation.

**Observation 1**. If $\Pi$ is a distinguishing feature of $T$ then for all featurettes $\tau$ present in $T$ it holds that $\Pi \cup \{\tau\}$ is a distinguishing feature of $T$.

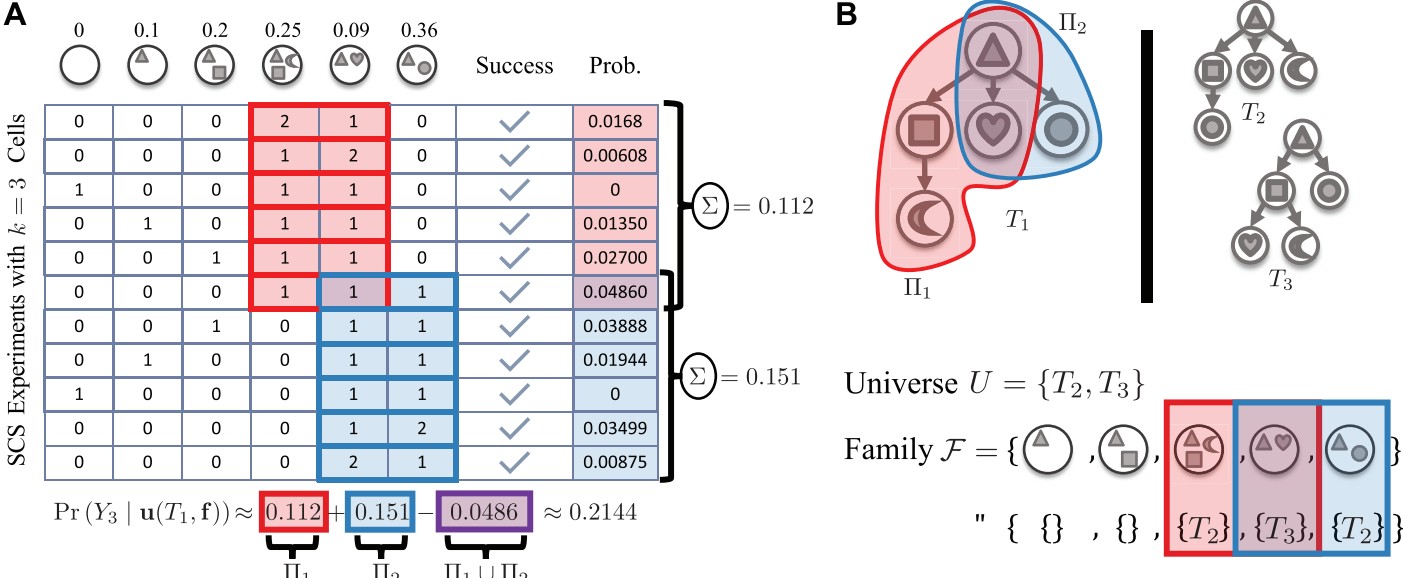

**Fig 3. PhyDOSE implementation details.** (A) To account for minimal distinguishing features that share featurettes, we use the inclusion-exclusion principle to compute $\Pr(Y_k|\mathbf{u}(T, \mathbf{f}))$. Here, $\Pi_1$ (red) and $\Pi_2$ (blue) share a featurette (with 'triangle' and 'heart' mutations). (B) To enumerate the set $\Phi^*$ of minimal distinguishing features of $T_1$, we reduce the problem to SET COVER and repeatedly identify minimum covers. Here, the universe $U$ is composed of trees $\{T_2, T_3\}$ and there is a subset in $\mathcal{F}$ for each featurette $\tau$ of $T_1$ composed of the trees where $\tau$ is absent.

This means that distinguishing features in $\Phi$ form a partially ordered set under the set inclusion relation. We call a distinguishing feature $\Pi$ *minimal* if there does not exist another distinguishing feature $\Pi' \in \Phi$ that is a proper subset of $\Pi$, i.e. $\Pi' \subsetneq \Pi$.

A direct consequence of Observation 1 is that the outcome of an SCS experiment is successful when we observe all featurettes of a distinguishing feature $\Pi$, and remains so even if we observe additional featurettes $\tau' \notin \Pi$.

As such, successful outcomes w.r.t. $\Phi$ equal those w.r.t. the set $\Phi^*$ of all minimal distinguishing features of $T$.

**Observation 2**. It holds that $\mathbf{c}(\Phi^*, k) = \mathbf{c}(\Phi, k)$.

Therefore, it suffices to restrict our attention to only $\Phi^*$ rather than the complete family $\Phi$ when computing $\Pr(Y_k | \mathbf{u}(T, \mathbf{f}))$ using (4). Section B.1 in S1 Text describes how to find $\Phi^*$ by reducing the problem to that of finding all minimal set covers, which we solve in an iterative fashion using integer linear programming.

## Power calculation for multiple biopsies

We now discuss the $T$-MUL-SCS-PC, which is the generalization of the $T$-SCS-PC problem to $b \geq 1$ biopsies. The key probability is $\Pr(Y_\mathbf{k} | U(T, F))$, i.e. the probability of concluding that $T$ is the true tree when sequencing $\mathbf{k} = [k_1, \ldots, k_b]^T$ cells from each biopsy. In the following, we discuss an exact (but computationally intensive) approach to compute this probability as well as a fast heuristic.

Given the numbers $\mathbf{k} = [k_1, \ldots, k_b]^T$ of cells to sequence from each biopsy, an *outcome* of the corresponding SCS experiment across $b$ biopsies is defined as a matrix $C = [c_{pi}]$ such that $c_{pi} \in \mathbb{N}$ is the number of cells that we observe from clone $i$ in biopsy $p$, and $\Sigma_{i=0} c_{pi} = k_p$ for all biopsies $p$. A *successful outcome* for a distinguishing feature $\Pi$ is an outcome where we observe at least one cell for each featurette in $\Pi$. Let $C(\Pi, \mathbf{k})$ denote the set of all successful outcomes for distinguishing feature $\Pi$. The set $C(\Phi^*, \mathbf{k})$ of successful outcomes for the minimal distinguishing feature family $\Phi^*$ is defined as the union $\bigcup_{\Pi \in \Phi^*} C(\Pi, \mathbf{k})$. Since sequencing of each biopsies proceeds independently, the probability of a observing an outcome $C = [\mathbf{c}_1, \ldots, \mathbf{c}_b]^T$ across $b$ biopsies equals

$$\Pr(\mathbf{c}_1, \ldots, \mathbf{c}_b \mid \mathbf{k}, U(T, F)) = \prod_{p=1}^{b} \Pr(\mathbf{c}_p \mid k_p, \mathbf{u}(T, \mathbf{f}_p)) = \prod_{p=1}^{b} \text{Mult}(\mathbf{c}_p \mid k_p, \mathbf{u}(T, \mathbf{f}_p)). \quad (5)$$

Hence, the desired tail probability of successful outcomes for the minimal distinguishing feature family $\Phi^*$ equals

$$\Pr(Y_\mathbf{k} \mid U(T, F)) = \sum_{[\mathbf{c}_1, \ldots, \mathbf{c}_b]^T \in C(\Phi^*, \mathbf{k})} \prod_{p=1}^{b} \text{Mult}(\mathbf{c}_p \mid k_p, u(T, \mathbf{f}_p)). \quad (6)$$

For a single biopsy ($b = 1$), the probability $\Pr(Y_\mathbf{k} | U(T, F))$ corresponds to sums of multinomial tail probabilities, enabling fast calculation using [26] as discussed in the previous section. This is no longer the case for $b > 1$ biopsies where the tail probability is over $b$ independent multinomial distributions (see product in Eq (6)). A naive way to compute $\Pr(Y_\mathbf{k} | U(T, F))$ would be to exhaustively enumerate all SCS outcomes with $\mathbf{k}$ cells, which scales exponentially in $\mathbf{k}$. As such, we develop a heuristic approach, which selects a subset of required featurettes/clones in each biopsy that together form a distinguishing feature in $\Phi^*$ and achieve the smallest total number of cells with confidence level $\gamma$. Section B.4 in S1 Text provides more details and Figure B in S1 Text provides an example where the heuristic returns a suboptimal solution.

## Consideration of SCS error rates

One current challenge with SCS is that the false negative rate per site is quite high with typical rates up to 0.4 for the commonly used multiple displacement amplification (MDA) method [32]. On the other hand, current false positive rates are low and are typically less than 0.0005 for MDA-based whole-genome amplification [32]. A *false negative* is defined as not observing a mutation that is present in the cell. A *false positive* occurs when we observe the presence of a mutation that did not occur in that cell.

With PhyDOSE, we propose one possible method for incorporating the false negative rate $\beta$ when it is known. Specifically, sampled cells follow a categorical distribution $\mathbf{u} = [u_0, \ldots, u_n]$ when conditioned on tree $T$. Hence, the probability of sampling a cell from clone $i$ equals $u_i$. True positives, i.e. correctly observing a mutation in a clone, follow a Bernoulli distribution with parameter $1 - \beta$. To observe a featurette/clone $i$ that has $n_i$ mutations and a prevalence of $u_i$, we thus need to have $n_i$ true positives. In other words, assuming independence among mutations, we require $n_i$ successful draws from a Bernoulli distribution parameterized by $1 - \beta$. As such, we derive new clonal prevalence $\mathbf{u}'(T, \mathbf{f}, \beta) = [u_i']$ from $\mathbf{u}(T, \mathbf{f}) = [u_i]$. Additionally assuming independence between the events of a cell being sampled from clone $i$ and the absence of false negatives, we set $u_i' = u_i(1 - \beta)^{n_i}$ where $n_i$ is the number of mutations in featurette/clone $i$. We set $u_0'$ to be equal to $1 - \sum_{i=1}^{n} u_i'$. This adjustment results in a reduction of the clonal prevalence and ultimately increases the value of $k^*$. The issue of false positives is less serious as error rates are low enough to be negligible.

## Prioritizing candidate trees post SCS experiment

The final step is to prioritize candidate trees after performing an SCS experiment with the number $k^*$ of cells computed by PhyDOSE. To this end, we compute the *support* of each tree $T \in \mathcal{T}$. Intuitively, support(T) is the number of cells that support the conclusion that $T$ is the actual phylogeny. Formally, we say that a distinguishing feature $\Pi$ of a tree $T$ is *observed* if each featurette of $\Pi$ is observed in at least one cell. Using this, we define support(T) as the number of cells that correspond to featurettes of an observed distinguishing feature $\Pi$ of $T$. Per Observation 1, it suffices to restrict our attention to the set $\Pi^*$ of minimal distinguishing features.

There are two outcomes of an SCS experiment with $k^*$ cells. Either there is no tree $T \in \mathcal{T}$ with non-zero support or there are one or more trees with non-zero support. In the former case, the SCS experiment has failed, which is expected to occur with probability $1 - \gamma$. In the latter case, which may occur in the presence of false negatives and false positives, we return the set of trees with maximum support.

Alternatively, we may use existing methods that infer tumor phylogeny from SCS data [14, 16, 33] or a combination of SCS and bulk data [19, 20].

## $k^*$ confidence interval

A common challenge in bulk sequencing is uncertainty in the cancer cell fractions $\mathbf{f}$ due to sampling of reads as well as sequencing and mapping errors. Following standard practice [9, 10, 34], we account for this uncertainty by taking confidence intervals $[\mathbf{f}^-, \mathbf{f}^+]$ as input. Typically, such confidence intervals are obtained by viewing variant read counts as draws from a binomial or beta-binomial distribution. Importantly, uncertainty in the cancer cell fraction leads to uncertainty in clonal prevalences. Therefore, for each tree $T \in \mathcal{T}$, we utilize $[\mathbf{f}^-, \mathbf{f}^+]$ to construct an interval $[k_-^*(T), k_+^*(T)]$ of the number of single cells reflecting the extreme values the clonal prevalences may assume. To find these values, we must consider the frequencies $[\mathbf{f}^-, \mathbf{f}^+]$ using

constraints from the tree $T$ following the sum condition (1) and the featurettes in a distinguishing feature of $T$. We do this using a heuristic that we describe in Section B.2 in S1 Text. We set the overall interval $[k_-^*, k_+^*]$ conservatively as the confidence interval $[k_-^*(T), k_+^*(T)]$ from the tree $T$ with the maximum $k_+^*(T)$ among all trees $T \in \mathcal{T}$.

### `phydoser` R package

We developed PhyDOSE and the associated optimizations into a freely available R package named `phydoser`. The functions in the `phydoser` R package are grouped into four areas: (i) I/O support (ii) pre and post processing (iii) PhyDOSE implementation iv) visualization. For I/O support, `phydoser` offers a suite of functions to read and convert external data into the data structures required by the R implementation of PhyDOSE. The pre and post processing capabilities include generation of the distinguishing feature families of each tree, the frequency inputs for the $k^*$ confidence interval and the computation of the support metric for a completed SCS experiment. The implementation of PhyDOSE includes both a single biopsy and a multiple biopsy mode. Lastly, the visualization functions facilitate creation of high resolution tree graphics while annotating the distinguishing feature or a specific featurette of a tree. `phydoser` is available at https://github.com/elkebir-group/phydoser.

## Results

In this section, we demonstrate the application of PhyDOSE to simulated and real data. We begin by validating our method using simulated data. Next, we provide retrospective results for a leukemia patient [23] and an acute myeloid leukemia cohort [27] where both bulk and single-cell DNA sequencing have been performed [23]. Finally, we use PhyDOSE to perform a prospective analysis to determine the required number of single cells to identify the true phylogeny in a non-small cell lung cancer patient cohort [3]. Source data and results can be found at https://github.com/elkebir-group/PhyDOSE.

### Simulations

**Design.** We used simulations to assess (i) the benefit of PhyDOSE's distinguishing feature analysis, (ii) robustness to uncertainty due to sequencing errors and (iii) robustness to violations of PhyDOSE's model assumptions. We generated simulated data where the ground truth tree $T^*$ is known. Given a fixed number $c$ of clones and $n$ mutations, we first generated a ground truth tree $T^*$ with $c$ vertices uniformly at random using Prüfer sequences [35] and randomly distributed the $n$ mutations to the $c$ clones while ensuring that every clone had at least one mutation. Next, we generated clonal prevalences $\mathbf{u} = [u_i]$ by drawing from a symmetric $(n + 1)$-dimensional Dirichlet distribution with concentration parameter 0.2. We used rejection sampling to ensure that each clonal prevalence $u_i$ was at least 0.05. Let $\sigma(i)$ be the set of clones that contain mutation $i$. We generated frequencies $\mathbf{f} = [f_i]$ by setting $f_i = \sum_{j \in \sigma(i)} u_j$ for each mutation $i \in \{1, \ldots, n\}$. We used the SPRUCE algorithm to enumerate the set $\mathcal{T}$ of trees given frequencies $\mathbf{f}$ [9].

To account for common single-cell sequencing errors, we varied false negative rates $\beta \in \{0, 0.2\}$ and doublet rates $\delta \in \{0, 0.1\}$. We generated, for each simulation instance, 10, 000 single cells sampled under the specified false negative rates $\beta$ and doublet rates $\delta$ according to the bulk clonal prevalence $\mathbf{u}$. To account for uncertainty in bulk sequencing, we additionally obtained confidence intervals on the cancer cell fractions of simulation instances sim3a, using a binomial distribution (with confidence $\alpha = 0.05$) and a mean coverage of 1000x (drawn from a Poisson distribution). Modeling additional uncertainty in bulk sequencing, sim4a instances

**Table 1. Simulation conditions.** We generate simulated data under eight conditions with 100 instances each. These conditions have varying subsets of candidate trees, number of clones, number of mutations per clone, clonal prevalence distortions, false negative and doublet rates. To analyze violations of the infinite sites assumption analysis, we introduced mutation losses in the sim1a instances. To analyze uncertainty in cancer cell fractions, we used the sim3a instances with a coverage of 1000x.

| ID | % of Trees | Clones | Mutations | Prevalence Noise | FNR $\beta$ | Doublet $\delta$ |
|---|---|---|---|---|---|---|
| sim1a | 100% | 7 | 7 | 0% | 0 | 0 |
| sim1b | 10% | 7 | 7 | 0% | 0 | 0 |
| sim2a | 100% | 7 | 7 | 5% | 0 | 0 |
| sim2b | 10% | 7 | 7 | 5% | 0 | 0 |
| sim2c | 100% | 7 | 7 | 20% | 0 | 0 |
| sim3a | 100% | 7 | 7 | 5% | 0.2 | 0.1 |
| sim3b | 10% | 7 | 7 | 5% | 0.2 | 0.1 |
| sim4a | 100% | 10 | 100 | 5% | 0.2 | 0.1 |

consist of $n = 100$ mutations each and use PyClone [36] to cluster the 100 mutations before enumerating $\mathcal{T}$.

Recall that PhyDOSE has three model assumptions: (i) ground truth tree $T^*$ is among the candidates trees $\mathcal{T}$, (ii) correspondence between clonal prevalences in bulk and subsequent single-cell sequencing samples, and (iii) infinite sites assumption for mutations. To assess (i), for simulation conditions 'b' (sim1b, sim2b and sim3b), we randomly sampled 10% of the trees outputted by SPRUCE [9]. To assess (ii), we varied single-cell clonal prevalences from the bulk clonal prevalences by resampling $\hat{\mathbf{u}} \sim \text{Dir}(\lambda \mathbf{u})$. We tuned the parameter $\lambda$ so that the clonal prevalence varied by an absolute average of 5% and 20% from the clones of the ground truth tree $T^*$, which resulted in $\lambda = 2000$ and $\lambda = 50$, respectively (Figure C in S1 Text). To assess (iii), we introduced violations of the infinite sites assumption (ISA) in the form of mutation losses. Specifically, we introduced one mutation loss in each of the instances of sim1a as follows. First, we randomly picked two distinct mutations $(i, j)$ where $i$ is introduced prior to $j$ in the ground truth tree $T^*$. Then, we designated the descendant mutation $j$ as a loss of mutation $i$ in each candidate tree $T \in \mathcal{T}$.

In total, we generated 100 simulation instances under eight varying conditions as specified in Table 1.

**The benefit of PhyDOSE's distinguishing feature analysis.** We compared PhyDOSE against SCOPIT [26], an existing method to design SCS experiments which takes as input a confidence level and the prevalence rate of each clone. Since SCOPIT does not connect the clones to be observed with a phylogenetic tree, we ran SCOPIT under two regimes. In the first regime (called 'SCOPIT'), we input the clonal prevalence rates for each $T \in \mathcal{T}$ and take the maximum SCOPIT output of all trees as an upper bound. In the second regime (called 'SCOPIT (true clones)'), we supplied SCOPIT with the prevalence rates of the simulated ground truth clones. The comparison to PhyDOSE was conducted at confidence level $\gamma = 0.95$.

PhyDOSE yielded a significant reduction in the number of cells to sequence compared to SCOPIT (Fig 4A). This is even the case when we provided the clonal prevalences of the ground truth tree to SCOPIT but not to PhyDOSE. In particular, for sim1a, SCOPIT required a median of 18.7 times as many cells than PhyDOSE, whereas SCOPIT (true clones) required 1.5 times as many cells (Fig 4A). In absolute numbers, PhyDOSE computed a median number of $k^* = 35$ cells compared to 544 cells computed by SCOPIT and 59 cells computed by SCOPIT (true clones) (Figure D in S1 Text).

To assess the accuracy of PhyDOSE's $k^*$ value, we generated follow-up *in silico* SCS experiments. Specifically, we ran our approach for prioritizing candidate trees and SPhyR [16] on sampled single cells. For the former, we performed 100 experiments for each simulation

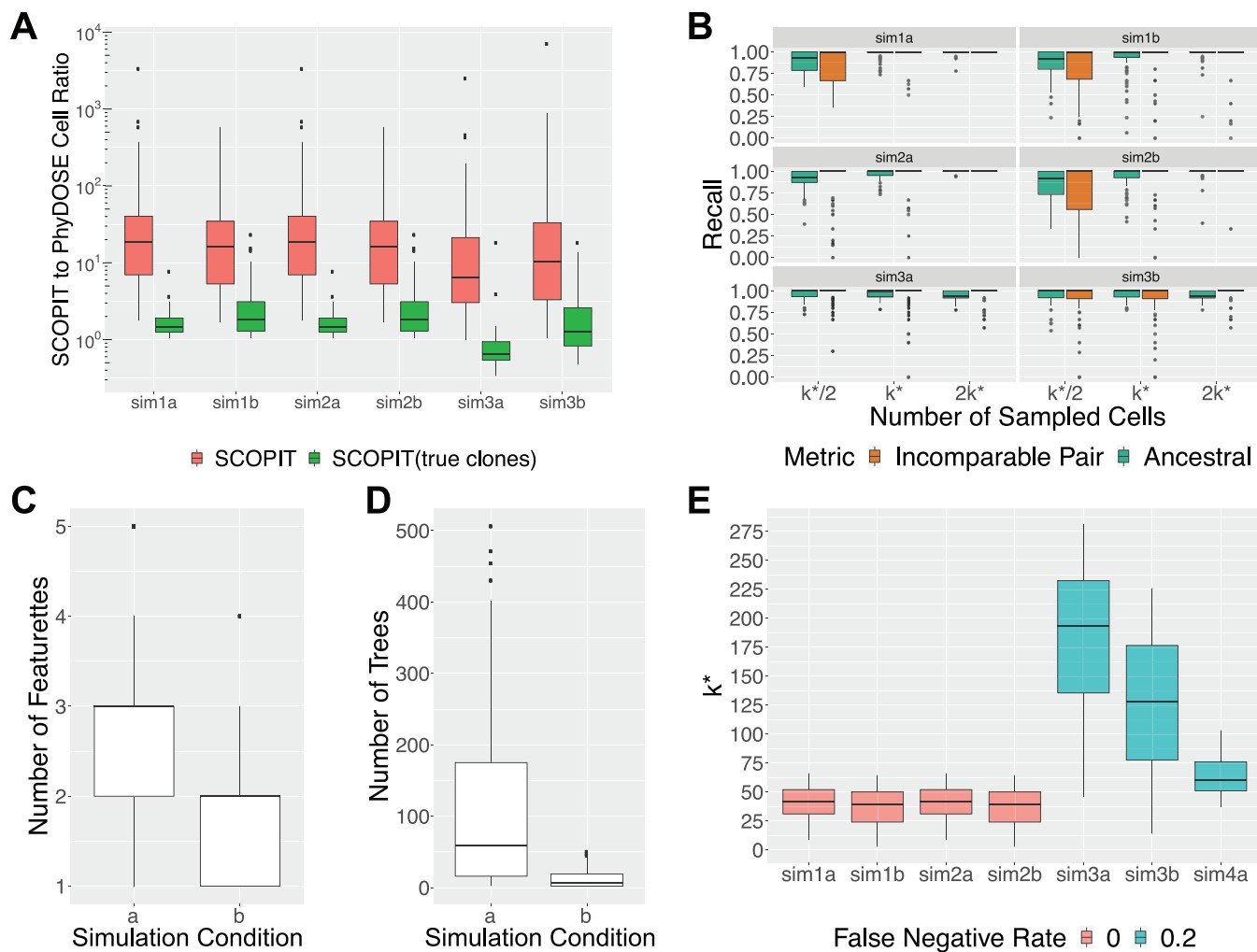

**Fig 4. Simulations demonstrate that PhyDOSE's calculated number of single cells resolves tree ambiguity in bulk sequencing data.** We used confidence level $\gamma = 0.95$ to determine the number $k^*$ of single cells to sequence. (A) SCOPIT to PhyDOSE cell ratio on a log scale when considering SCOPIT in a worst case regime where the true phylogeny is unknown and a best case regime where SCOPIT utilizes the clonal prevalance of the clones in the simulated ground truth tree. (B) Recall metrics of the tree inferred by SPhyR [16] by randomly sampling $k^*/2$, $k^*$ and $2k^*$ simulated single cells. (C) Number $|\Pi|$ of featurettes among minimal distinguishing features $\Phi^*$ when compared between the enumerated candidate set $\mathcal{T}$ (conditions $a$) and the downsampled candidate set (conditions $b$). (D) Number $|\mathcal{T}|$ of trees in the candidate set when enumerated by SPRUCE [9] (condition $a$) and when downsampling the enumerated candidate set (condition $b$). (E) Number $k^*$ of cells identified by PhyDOSE.

instance, reporting the number of experiments that successfully recovered the ground truth tree $T^*$. We counted an experiment as successful if we correctly and uniquely selected the ground truth tree as $T^*$. For the latter, we also considered the performance of SPhyR when sampling half and double the number $k^*$ of cells determined by PhyDOSE.

Figure F in S1 Text shows that the prioritization approach worked particularly well for sim1a with a median success rate of 96%. Similarly, SPhyR [16] was able to identify the true tree in the majority of cases after sampling PhyDOSE computed number $k^*$ of cells. To quantify the similarity between the tree $T$ estimated by SPhyR and the true tree $T^*$, we used two commonly-used tree distance metrics, ancestral and incomparable pair recall. *Ancestral pair recall* is defined as $|\mathcal{A}(T) \cap \mathcal{A}(T^*)|/|\mathcal{A}(T^*)|$ where $\mathcal{A}(T)$ ($\mathcal{A}(T^*)$) is the set of ordered pairs of mutations that occur on distinct edges of the same branch of $T$ ($T^*$). *Incomparable pair recall* is defined as $|\mathcal{I}(T) \cap \mathcal{I}(T^*)|/|\mathcal{I}(T^*)|$ where $\mathcal{I}(T)$ ($\mathcal{I}(T^*)$) is the set of unordered pairs of

mutation that occur on edges in distinct branches in $T$ ($T^*$). For sim1a, the median of both metrics is 1 when sampling $k^*$ cells, reflecting that SPhyR identified the true tree in the majority of cases (Fig 4B). Moreover, we found greater gains in performance between sampling $k^*$ cells versus $k^*/2$ cells than sampling $2k^*$ cells versus $k^*$ cells.

Fig 4C shows the number of clones in each minimal distinguishing feature identified by PhyDOSE, ranging from 1 to 5 with a median of 3 for simulation conditions 'a'. Importantly, this number is smaller than the total number of 7 clones, demonstrating that a distinguishing feature yields an efficient representation of a tree. This led to a smaller number $k^*$ of cells inferred by PhyDOSE compared to SCOPIT without sacrificing performance in the tree reconstruction from the follow-up SCS experiment.

**Robustness to violations of PhyDOSE's model assumptions.** We used simulations to assess PhyDOSE's performance when model assumptions are violated. We begin with the case where the ground truth tree $T^*$ is not guaranteed to be present among the candidate trees $\mathcal{T}$. In sim1b, we downsampled the set of candidate trees of the instances in sim1a to 10% (Fig 4D). Similarly to sim1a, PhyDOSE significantly reduced the required number $k^*$ of single cells compared to SCOPIT (Fig 4A). We analyzed follow-up *in silico* SCS experiments using the prioritization approach and SPhyR [16]. Since the prioritization approach only considers candidate trees $\mathcal{T}$, which may not contain the ground truth tree, it is not surprising that performance dropped to a median success rate of 0%. However, SPhyR was able to identify the ground truth tree in the majority of cases (median of 1 for both incomparable and ancestral pair recall).

Next, we assessed the impact of clonal prevalence distortions between bulk and single cell data in sim2a, sim2b and sim2c. We found PhyDOSE to be robust to random clonal prevalence noise between bulk and single-cell sequencing as evidenced by a drop of only 1% in the median percentage of successful *in silico* SCS experiment when there is no downsampling of trees (Figure C in S1 Text). Additionally, the recall performance metrics (Fig 4B, Figure C in S1 Text) are also not substantially different between sim1a versus sim2a and sim2c, showing similar trends when using $k^*/2$ and $2k^*$ cells. We attribute this to the fact that PhyDOSE's use of distinguishing features relies on the clonal prevalence of a few key clones. Furthermore, PhyDOSE performs well for the case where candidate trees have been downsampled and clonal prevalences have been distorted (sim2b, see Fig 4B).

Finally, we assessed the sensitivity of PhyDOSE to ISA violations by applying it to a candidate set $\mathcal{T}$ of 1-Dollo phylogenies obtained from sim1a. Specifically, the resulting simulation instances had candidate trees composed of 6 mutations, one of which having undergone a loss. We performed 100 *in silico* experiments for each of the 100 simulation instances. Like sim1a under the ISA, the median percentage of successful experiments is 95% for the 1-Dollo phylogenies (Figure H in S1 Text). However, 23 simulation instances had a success rate of 0%. This increased variance is to be expected as distinguishing features may no longer be distinguishable from those of other trees when mutation losses occur. Nevertheless, PhyDOSE's suggested SCS experiments for these 23 instances significantly reduced the number of candidate trees from a median of 161 trees pre-experiment to a median of 5 trees post-experiment (Figure H in S1 Text). In all cases, the ground truth tree was included in the candidate set after performing the SCS experiment. Hence, as distinguishing features provide an efficient representation for each tree using only a subset of clones, PhyDOSE performed well on data with violations of the infinite sites assumption.

**Robustness to uncertainty due to sequencing errors.** Sequencing errors occur in both the initial bulk sequencing experiment as well as the follow-up single-cell sequencing experiment. We begin by considering common SCS errors using false negative rate $\beta = 0.2$ and doublet rate $\delta = 0.1$ in sim3a and sim3b. While the number $k^*$ significantly increased compared to

the simulations without false negatives (Fig 4E), PhyDOSE's computed number $k^*$ of cells remained an order of magnitude smaller than SCOPIT but not for SCOPIT (true clones) (Fig 4). The latter is due to PhyDOSE's adjustments of the clonal prevalence when factoring in a false negative rate of $\beta = 0.2$ in trees other than $T^*$ (Figure E in S1 Text). Tree inference using SPhyR in subsequent SCS experiments based on PhyDOSE's $k^*$ identified the ground truth tree in the majority of cases (median of 1 for both incomparable and ancestral pair recall).

Figure G in S1 Text shows PhyDOSE performance in sim4a instances with mutation clusters inferred by PyClone [36]. We included the clustered pair recall in our analysis defined as $|\mathcal{C}(T) \cap \mathcal{C}(T^*)|/|\mathcal{C}(T^*)|$, where $\mathcal{C}(T)$ ($\mathcal{C}(T^*)$) is defined as the set of unordered pairs of mutations that are introduced on the same edge in $T$ ($T^*$). At $k^*$ cells, the median ancestral pair recall was 0.96, the incomparable pair recall was 0.86 and the clustered pair recall was 0.94, showing a reduction in performance from the first sim1, sim2 and sim3 simulations due to the additional errors introduced by PyClone [36].

Finally, we evaluated the performance of PhyDOSE in the presence of cancer cell fraction uncertainty by adapting sim3a instances. We simulated variant and total read counts with a coverage of 1000x. Then, we constructed a binomial proportion 95%-confidence interval using Jeffrey's prior interval [37]. Next, we used the mean of this interval to enumerate the candidate set of trees with SPRUCE [10]. Using the method described in Section B.2 in S1 Text, we constructed a confidence interval $[k^*_-, k^*_+]$ for each replication (Figure I in S1 Text). We performed 100 in silico for each replication using both $k^*_-$ and $k^*_+$, selecting the tree $T \in \mathcal{T}$ with the maximum support($T$) as $T^*$. We obtained a median percentage of successful trials of 86% (IQR: 44%–93%) and 86% (IQR: 31%–98%) when randomly sampling $k^*_-$ and $k^*_+$ in silico cells, respectively (Figure I in S1 Text).

**Running time.**   We performed an empirical run time analysis on a server with two Intel Xeon Gold 5120 CPUs @ 2.20GHz and 512 GB RAM. Performing the power calculation of $k^*$ is fast [26] when $|\Phi| = 1$ and the median of $|\Phi|$ in our simulations was 1. Therefore, the main bottleneck in PhyDOSE is the determination of $\Phi$ for each tree in the candidate using the algorithm presented in Section B.1 in S1 Text. Note that this step is embarrassingly parallelized because $\Phi$ is computed independently for each $T \in \mathcal{T}$. However, we leave the parallel `phydoser` implementation as future work. To additionally explore how PhyDOSE scales with $|\mathcal{T}|$, we generated an additional simulation set with 10 mutations and no mutations clusters and set a ten minute time limit on finding the distinguishing features. We calculated the runtime in seconds for each simulation instance when $\Phi$ is solved sequentially for each $T \in \mathcal{T}$. The largest input size to complete was 9901 required 619 seconds with 92% of the the spent on finding the distinguishing features. The results are displayed in Figure J in S1 Text.

In summary, our simulations demonstrate that PhyDOSE's distinguishing feature analysis results in significantly fewer cells to sequence than SCOPIT [26] without a subsequent loss in power to identify the true phylogeny. Moreover, we find that PhyDOSE is robust to typical sequencing errors in both the bulk and SCS data as well as violations of model assumptions.

## Retrospective analysis of an acute lymphoblastic leukemia patient

We considered a cohort of six childhood acute lymphoblastic leukemia (ALL) patients whose blood was sequenced using bulk and targeted single-cell DNA sequencing [23]. The number of sequenced single cells per patient varied between 96 and 150. To validate our approach, we used PhyDOSE to calculate the number $k(T^*)$ of cells needed to identify the true phylogeny $T^*$ that is consistent with both data types, thereby retrospectively determining whether fewer single cells suffice to determine $T^*$, decreasing the cost of replicate experiments. In addition, we assessed whether the calculated number $k(T^*)$ yielded $T^*$ using in silico SCS experiments.

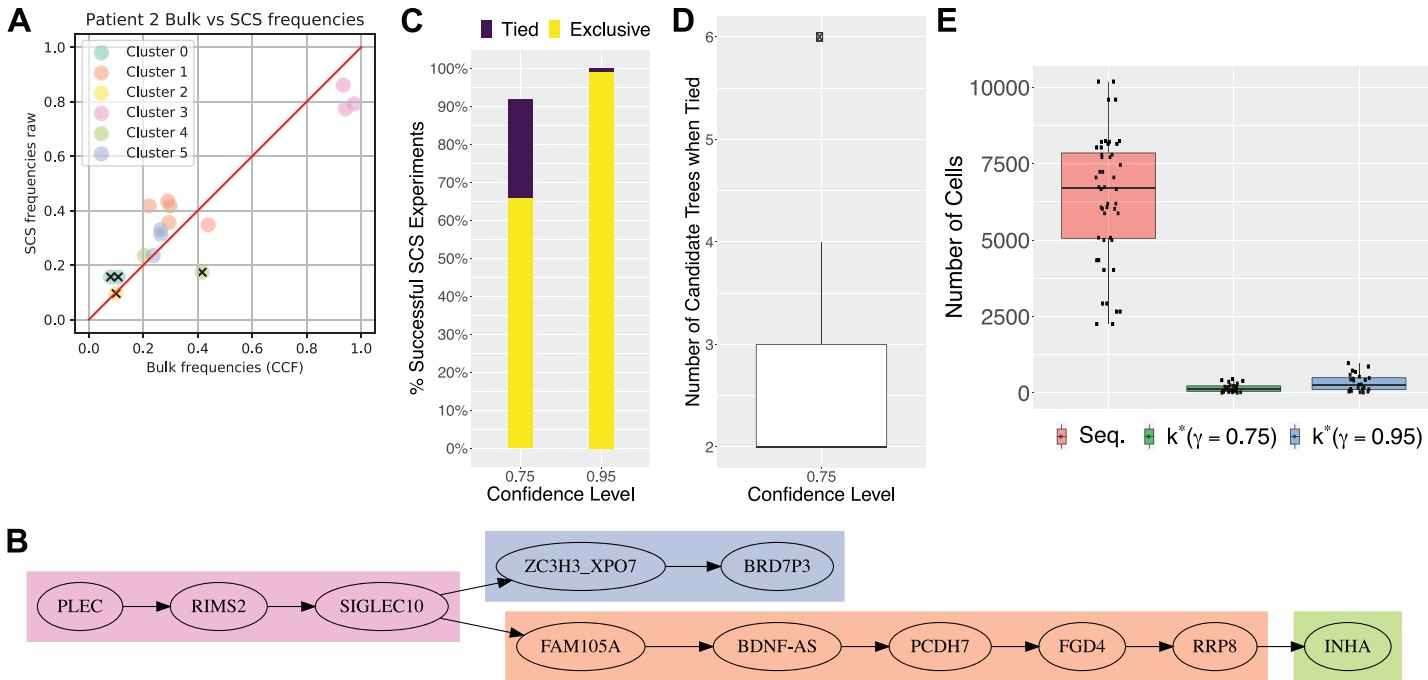

**Fig 5. Retrospective analysis of ALL patient 2 [23] and AML cohort [27] demonstrates that fewer cells suffice for replication.** Panels (A)-(D) consider ALL patient 2 [23] and panel (E) considers the AML cohort [27]. (A) There is a strong correlation between bulk and single-cell mutation frequencies. Colors indicate mutation clusters from SCS data and excluded mutations are indicated by 'x'. (B) Phylogeny $T^*$ that is consistent with the SCS and bulk data. (C) Percent of successful outcomes in 100 *in silico* SCS experiments, obtained by sampling from the 115 sequenced cells without replacement following PhyDOSE's calculated number $k(T^*)$ of cells (103 for $\gamma = 0.95$ and 50 for $\gamma = 0.75$). Exclusive outcomes (yellow) uniquely identified $T^*$ whereas tied outcomes (purple) yielded a small set of candidate phylogenies that include $T^*$. (D) Number of candidate phylogenies in the case of ties. (E) The distribution of PhyDOSE's $k^*$ for $\gamma \in \{0.75, 0.95\}$ of all patients in the AML cohort with $|\mathcal{T}| > 2$ as well as the number of cells that were originally sequenced.

Due to the absence of published copy-number aberration information for this dataset, we focused our attention on patient 2 whose single-cell phylogeny adhered to the infinite sites assumption and the variant allele frequencies suggested the absence of copy-number aberrations (as detailed in Section C in S1 Text). For this patient, 16 autosomal mutations in 115 cells were sequenced [23]. We note that the authors had no knowledge of the number of cells that would suffice to infer the tumor phylogeny of the patient. Using the infinite sites assumption and assuming the absence of copy-number aberrations, we define the cancer cell fraction, or frequency $f_i$ of each mutation $i$ in the bulk data as $2 \cdot \text{VAF}(i)$. We define the *SCS mutation frequency* as the fraction of single cells that harbor the mutation. Strikingly, there is a clear correlation between the bulk and SCS mutation frequencies, supporting PhyDOSE's first assumption (Fig 5A). We excluded mutation *CMTM8* because of a notable discrepancy in frequencies (0.4 in bulk vs. 0.2 in SCS). Using SPRUCE [9], we enumerated the set $\mathcal{T}$ of trees from the bulk data, yielding over 2.5 million trees. This number is mainly driven by 3 mutations (*ATRNL1*, *LINC00052* and *TRRAP*) with a VAF less than 0.05. Excluding these 3 mutations resulted in a more tractable number of 2, 576 trees. We note that in practice we may similarly exclude mutations because of very low VAFs or less importance in downstream analyses. Fig 5B shows the single tree $T^* \in \mathcal{T}$ that was consistent with the cleaned single-cell data, supporting PhyDOSE's second assumption.

We ran PhyDOSE using varying confidence levels $\gamma \in \{0.75, 0.95\}$ and an estimated false negative rate of $\beta = 0.2$ reported by the authors [23]. PhyDOSE calculated that $k(T^*) = 103$ cells suffice to identify $T^*$ with confidence level $\gamma = 0.95$. Indeed, performing 100 *in silico* SCS

experiments, by sampling $k(T^*)$ cells among the 115 sequenced cells without replacement, yielded a success rate of 99% (Fig 5C).

To reduce costs, we explored what would have happened retrospectively with a lower confidence level $\gamma$ of 0.75. PhyDOSE calculated that $k(T^*) = 50$ cells are needed for $\gamma = 0.75$, which is a significant cost savings over $\gamma = 0.95$. Performing 100 *in silico* SCS experiments yielded a success rate of uniquely identifying $T^*$ of 66%, which was lower than the expected rate of 75%. Furthermore, we noted that in an additional 26% of experiments the correct phylogeny $T^*$ was among the trees with the highest overall support (Fig 5C). The number of trees in the tied set of successes varied from 2 to 6 (Figure L in S1 Text), showing that although PhyDOSE did not uniquely identify the tree, it was able to significantly reduce the original set of 2576 trees (Figure L in S1 Text and Figure M in S1 Text).

In summary, this retrospective analysis shows that the true tree for patient 2 could have been identified confidently with fewer cells than the 115 cells initially sequenced [23]. With a lower confidence level $\gamma$, PhyDOSE computes that far fewer cells are required, significantly reducing costs but at the expense of a lower success rate of uniquely identifying the true phylogeny. Nevertheless, the resulting SCS experiment will eliminate a large fraction of the original set of candidate phylogenies due to the incorporation of distinguishing features in the PhyDOSE power calculation.

## Retrospective analysis of an acute myeloid leukemia cohort

Morita et al. [27] performed high-throughput targeted microfluidic single-sequencing using the Tapestri platform [38] on a cohort of 77 patients with acute myeloid leukemia (AML). The authors additionally performed bulk sequencing in order to confirm the presence of a mutation in the single-cell data. We note that the authors restricted their analysis to somatic mutations (SNVs and indels) that did not occur in regions affected by additional copy number aberrations.

Here, we utilized the published bulk sequencing VAFs of the SNVs in each patient, eliminating any mutations not detected via bulk sequencing, to enumerate a set of candidate trees using SPRUCE [9]. We restrict our analysis to the 24 patients where bulk sequencing data was available and SPRUCE identified more than one candidate tree. The median number of mutations for these patients was 4 (IQR: 3-5). We retrospectively used PhyDOSE at confidence levels $\gamma \in \{0.75, 0.95\}$ to estimate the cells needed to perform an equivalent single-cell experiment. We used false negative rate $\beta = 0.049$, which is the mean of the per patient published false negative rate. In the original study, a median of 7, 584 cells per patient (IQR: 6, 194–8, 361) were sequenced. Fig 5E shows the distribution of PhyDOSE $k^*$ for the 24 patients (median is 2, IQR: 2–6, max is 316) at $\gamma \in \{0.75, 0.95\}$ versus the total number of cells sequenced in [27]. For $\gamma = 0.95$, the median value of $k^*$ was 274 cells (IQR: 230–497). This is a significant reduction from the number of cells sequenced per patient in [27] with a median percent reduction at confidence level $\gamma = 0.95$ of 95.4% (IQR: 92.2%–98.0%) (Table A in S1 Text).

For these 24 patients, Morita et al. [27] sequenced 153, 558 cells while the PhyDOSE design at confidence level $\gamma = 0.95$ requires 8,144 cells (Table A in S1 Text). Using that a Tapestri run of 10,000 cells costs $795 and including additional sequencing costs of $200 per run with NovaSeq or $1000 per run with MiSeq [39, 40], we estimate the costs of the original study as $15,920 and the estimated costs of the PhyDOSE design as $1,995. This assumes the original study utilized 16 Tapestri runs with NovaSeq while PhyDOSE requires 1 Tapestri run with the more expensive MiSeq to avoid multiplexing. Thus, designing the experiment with PhyDOSE would have yielded a 93.75% cost savings. Further, we note that this study design requires

targeted sequencing and that the potential cost savings when the experimental design requires whole genome sequencing would further increase.

### Prospective analysis of a non-small cell lung cancer cohort

Using PhyDOSE, we prospectively determined the number of cells needed to uniquely identify the true phylogeny for the 25 out of 100 patients in the TRACERx non-small-cell lung cancer cohort that have multiple candidate trees [3]. The authors previously identified the set $\mathcal{T}$ of candidate trees for each patient using CITUP [11] after clustering mutations with PyClone [36]. The authors also reported the cancer cell fraction of each mutation cluster in each bulk sample. The number of trees in the candidate set for each patient ranged from 2 to 17, with each containing mutation clusters with between 5 and 882 mutations (Table B in S1 Text).

Assuming high confidence on the co-occurrence of mutations in a cluster, mutation clusters alleviate the issue of false negatives, i.e. it suffices to only observe a small number of mutations to impute the presence of the other mutations in the same cluster. Here, with a typical SCS false negative rate of 0.2, the probability of all mutations in the smallest cluster (with size 5) dropping out thus equals $0.2^5 = 0.00032$, a probability that can be neglected. As such, we set $\beta = 0$. Unlike in the simulations and the previous real datasets, multiple bulk samples corresponding to distinct spatial locations were available for analysis per patient. In addition to the naive method where we select a single biopsy that minimizes $k^*$, we used the multiple biopsy heuristic to infer numbers $\mathbf{k}^*$ of cells for each biopsy. For both methods, we used confidence level $\gamma = 0.95$.

Following the naive approach, PhyDOSE returned a finite value of $k^*$ for 24 out of the 25 patients. The naive approach yielded $k^* = \infty$ for patient CRUK0037 because for each of the 5 biopsies there is a tree where every distinguishing feature is not observable. That is, the clonal prevalence of one of the comprising featurettes is 0. By contrast, the heuristic calculated a total of 243 cells (R1: 36; R2: 49; R3: 60; R4: 38 and R5: 51 cells) for this patient. For patients CRUK0013 and CRUK0076, the naive approach required the sequencing of more cells from a single biopsy than the multiple biopsy heuristic (CRUK0013: 1, 051 vs. 215 cells; CRUK0076 47, 479 vs. 48 cells). For the remaining 22 patients, treating the samples independently yields the same number of cells from the selected biopsy as the heuristic. Table B in S1 Text and Fig 6 provide detailed numbers.

These strikingly low values of total number of cells for the 25 patients with multiple candidate trees and multiple biopsies demonstrate the benefit of using PhyDOSE to strategically optimize the design of follow-up single cell experiments.

### Discussion

In this work, we showed that the mutation frequencies $\mathbf{f}$ and the set $\mathcal{T}$ of tumor phylogenies inferred from initial bulk data contain valuable information to provide guidance for follow-up SCS experiments. We introduced PhyDOSE, a method to calculate the number $k^*$ of single cells needed to infer the true phylogeny $T^*$ given $\mathbf{f}$, $\mathcal{T}$ and a user-specified confidence level $\gamma$. Underpinning our method is the observation that often only a subset of clones suffices to distinguish one tree $T \in \mathcal{T}$ from the remaining trees $\mathcal{T} \setminus \{T\}$. Although PhyDOSE is motivated by the output of deconvolution methods for bulk sequencing, it is agnostic to the method used to obtain the candidate set as long as the clonal prevalence rates of the distinguishing features can be estimated. Thus, the input set $\mathcal{T}$ of candidate trees can be obtained from preliminary single-cell and/or bulk sequencing data. Similarly, PhyDOSE is agnostic to the phylogeny inference method used to analyze data from the proposed SCS experiment. We also provided

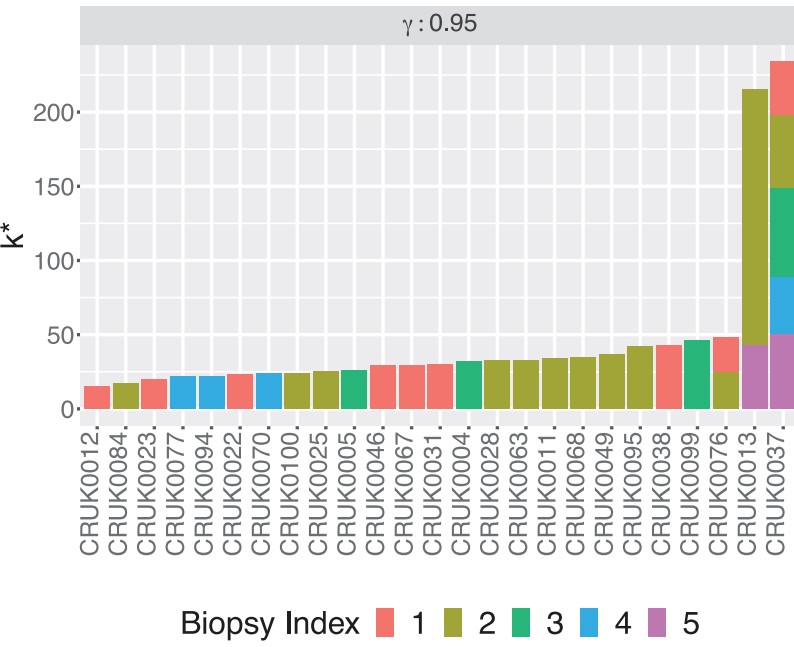

**Fig 6. PhyDOSE multiple biopsy heuristic calculated numbers k* of cells per biopsy for the lung cancer cohort [3] at confidence level γ = 0.95.**

heuristics for realistic scenarios that arise in practice, such as handling uncertainty in the estimation of cancer cell fractions and the availability of multiple biopsies.

We validated PhyDOSE using simulations and a retrospective analysis of leukemia patients [23, 27], concluding that PhyDOSE's computed number $k^*$ of cells resolves tree ambiguity, even in the presence of SCS errors. Our simulations showed that PhyDOSE remains robust in the presence of sequencing errors and violations of model assumptions, outperforming the competing method, SCOPIT [26]. In a prospective analysis, we demonstrated that only a small number of cells suffice to disambiguate the solution space of trees in a recent non-small cell lung cancer cohort [3]. In summary, PhyDOSE proposes cost-efficient SCS experiments that will yield high-fidelity phylogenies, which may consequently improve downstream analyses in cancer genomics aimed at deepening our understanding of cancer biology.

There are several future research directions. First, in the case of multiple bulk samples, although we propose an exact calculation, we only implement a heuristic since the exact calculation does not scale to realistic problem sizes. Developing an implementation of the exact calculation in the case of multiple samples would yield a further cost reduction in the experimental design since the heuristic overestimates the number of cells at given confidence level $\gamma$. Second, to further reduce SCS costs, we might want to include a mutation selection step as part of our approach to perform targeted rather than whole-genome sequencing. Third, similar ideas can be used to design follow-up sequencing experiments using alternative sequencing technologies such as long read sequencing. Alternatively, performing additional bulk sequencing rather than single-cell sequencing might be more cost-effective, especially when obtaining a bulk sample with distinct clonal prevalences [10, 41]. Fourth, we plan to develop an easy-to-use Shiny user interface to facilitate the use of PhyDOSE for the design of sequencing experiments. Fifth, to improve robustness in the presence of SCS errors, we plan to explore alternative definitions of successful SCS experiment outcomes, requiring that more than one cells is observed of each featurette of a distinguishing feature. This will enable us to

address errors such as doublets and false positives in an SCS experiment. Similar ideas can be used to address uncertainty in mutation clusters inferred from bulk sequencing data. Sixth, the concept of distinguishing features may be useful to summarize diverse solution spaces in cancer phylogenetics [42]. Finally, we plan to explore evolutionary models beyond the infinite sites model, such as the Dollo parsimony model where mutations might be lost [16], requiring a more careful approach to find the distinguishing features of a tree.

## Supporting information

**S1 Text. Supplementary materials.**
(PDF)

## Author Contributions

**Conceptualization:** Mohammed El-Kebir.

**Data curation:** Leah L. Weber, Nuraini Aguse.

**Formal analysis:** Leah L. Weber, Nuraini Aguse.

**Funding acquisition:** Nicholas Chia, Mohammed El-Kebir.

**Investigation:** Leah L. Weber, Nuraini Aguse, Mohammed El-Kebir.

**Methodology:** Leah L. Weber, Nuraini Aguse, Mohammed El-Kebir.

**Project administration:** Mohammed El-Kebir.

**Resources:** Mohammed El-Kebir.

**Software:** Leah L. Weber, Nuraini Aguse.

**Supervision:** Mohammed El-Kebir.

**Validation:** Nicholas Chia.

**Visualization:** Leah L. Weber, Nuraini Aguse.

**Writing – original draft:** Leah L. Weber, Nuraini Aguse, Mohammed El-Kebir.

**Writing – review & editing:** Leah L. Weber, Nuraini Aguse, Nicholas Chia, Mohammed El-Kebir.

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
