## [Decision Letter · Decision Letter 0]

1 Jun 2020

Dear Dr. El-Kebir,

Thank you very much for submitting your manuscript "PhyDOSE: Design of Follow-up Single-cell Sequencing Experiments of Tumors" for consideration at PLOS Computational Biology.

As with all papers reviewed by the journal, your manuscript was reviewed by members of the editorial board and by several independent reviewers. In light of the reviews (below this email), we would like to invite the resubmission of a significantly-revised version that takes into account the reviewers' comments.

While the reviewers acknowledge that this work represents a rigorous methodological contribution to this field, they raise concerns about comparisons with other methods (reviewer #3) and practical limitations (reviewer #1, #2) that need to be addressed in a revised manuscript.

We cannot make any decision about publication until we have seen the revised manuscript and your response to the reviewers' comments. Your revised manuscript is also likely to be sent to reviewers for further evaluation.

Sincerely,

Florian Markowetz

Deputy Editor

PLOS Computational Biology

Florian Markowetz

Deputy Editor

PLOS Computational Biology

While the reviewers acknowledge that this work represents a rigorous methodological contribution to this field, they raise concerns about comparisons with other methods (reviewer #3) and practical limitations (reviewer #1, #2) that need to be addressed in a revised manuscript.

Reviewer's Responses to Questions

**Comments to the Authors:**

Reviewer #1: This paper develops a method, PhyDOSE, for designing follow-up single-cell sequencing experiments to resolve tumor phylogenies based on initial bulk sequence analysis. This paper is based on a conference submission that has already been through a round of revision in response to the conference reviewers. Although I was not one of the reviewers of the conference version, I recognize that it is more polished than a typical new submission as a result and that the conference reviewers already raised what would have been some of my concerns about the paper. I therefore read it in the spirit of a paper that has already been through a round of revisions. The general problem is well motivated. Although the cost of single-cell sequencing is dropping rapidly, it remains costly for large studies and there is good reason to use it efficiently. The approach proposed here appears sensible and technically sound, making good use of prior theory of the authors on tumor phylogeny enumeration, adding robust handling for common kinds of data errors in single-cell sequencing, and connecting the theory nicely to a rigorous new probabilistic model for use in power calculations. The method is well backed up by empirical analysis of simulated data and three real cohorts, showing the method to be effective and to improve substantially on naïve approaches in at least some cases. Some natural concerns about the approach in the conference version, such as robustness to biased sampling or to failing to identify some clones, were raised by the conference reviewers and effectively rebutted by new experiments. The method does still make some restrictive assumptions and limitations, as was pointed out by the prior reviewers. While these are not fully resolved even in the revision, the authors do have fair responses for them.

Given this, I have little to add in the way of criticism and would consider my remaining points largely discretionary. My only real substantive concern is that some of the limitations of the method raised in the conference reviews are still limitations and one might question whether it is sufficient in some cases to note them and defer them to future work. I refer here essentially to the points raised in the final paragraph of Discussion.

In that regard, the use of the infinite sites model is questionable enough that one could argue it needs to be at least demonstrated that the method is reasonably robust to violations. While many methods in this space use the infinite sites assumption, it is well established that it is not consistently accurate and is at least becoming more accepted that methods must handle some violations. I think it is fine to defer to future work extension to a more robust model like Dollo parsimony, which would understandably require some significant changes to the theory and algorithms, so long as the method works reasonably well on data that violates the assumption without that.

The paper also considers a criticism about the use of single rather than multiple bulk samples and defers that question to future work. There is good evidence that phylogeny inference from single bulk samples is simply not accurate enough to be the basis for even the initial step of a combined bulk and single-cell study, and so one might reasonably argue that accommodating multiple bulk samples is so important that it should be part of even a first method of this class.

I will also just raise as a discretionary thought some other possible scenarios where I could imagine this method being useful. I wonder if the method could be applied if there has already been some bulk and some single-cell sequencing done, as in some studies to date, and we want to plan further single-cell sequencing. Would the method be adaptable to such a case? Or could it do better if we assume multiple batches of single-cell sequencing, with an opportunity to reevaluate after each batch? I can accept that these are getting far enough afield that they do not need to be solved in this paper, but might also be questions for future work.

Reviewer #2: This paper discusses PhyDOSE, a method to perform power calculation for single-cell sequencing, when we need to disentangle the clone tree associated to a tumour sample. The idea is that, while we often perform a bulk sequencing experiment to assess a number of possible trees that fit the mutation allele frequency (VAF), it happens often that more than one tree are equally-likely to fit the data. If we can generate single-cell sequencing data of a number X of single-cells, then we can disambuiguate which tree best fits the data. PhyDOSE is a method that tells us what should be the value of X to bound the probability that we can determine a unique best tree.

The paper is clear, and the problem is known in the field. There is abundant literature explaining/ showing that determining a single tree from bulk data can be challenging, therefore the solution of sequencing single cells can be appealing, as much as other approaches. The ILP formulation of the problem seems to be correct, and the results and methods consistent with the theory.

However, there are some major limitations of the work in the current form. I think fixing them would make the main message (a computational design technology) appealing.

- I do not think that you can prescind from the fact that many datasets collect multiple tumour bulks at once, as you also note in your Discussion. This requires a multivariate problem definition, in principle, that you need at least to discuss. There are multi-region sequencing simulators that you can use to this respect, if you want to try to simualte data. The current work presents instead an independence assumption, and uses that in the current analyses (sect "Prospective Analysis of a Non-small Cell Lung Cancer Cohort"). The choice of PhyDOSE is to minimise the number of cell estimates across all samples; is this supported by some consideration?

- [related to the above] what do the author mean by saying that "Mutation clusters alleviate the issue of false negatives, i.e. it suffices to only observe a single mutation to impute the presence of the other mutations in the same cluster.". Imputation can be tricky; if I observe a low-VAF mutation in 3 out of 4 biopsies, I think that the imputation should depend on the coverage at the locus. If high-enough, imputation can be supported by a statistical argument based on Binomial testing on read counts (what are the odds of not-seeing a mutation with a certain VAF with my current coverage). If low, imputation might generate false positives. Is it possible to frame this uncertainty in PhyDOSE's computation of the optimal number of cells for these scenarios?

- You present some reduction in cells numbers that are not exceptionally striking. Can you justify a difference in sequencing cost for the effort of using your design method? At the end of the day, if one does not save a substantial amount of sequencing costs, why would he/ she bother using PhyDOSE? I think you need to provide stronger evidence of why your computations can be important for a molecular biologist that is designing a new experiment. If the reduction is not substantial, I think that your contribution would be just theoretical and could be less appealing. In the context of sequencing technologies, you should put effort to understand the cost for standard experimental setups (e.g., I presume you would be using either a deep sequencing panel, or a digital-PCR assay) and their possible parametrisation. On your real data you can effectively discuss these reductions (assuming certain costs since you did not generate the data).

- It is increasingly evident that a number of "clusters" identified through standard VAF deconvolution method can represent random ancestors constituted by neutrally evolving mutations (https://doi.org/10.1101/586560). Clustering tail mutations is also wrong because tail lineages are polyphyletic. You should discuss this when you consider the problem of using certain subsets of mutations. Since some of the input clusters should be removed from the clone trees, and you could discuss what happens if you end up taking cells from those clusters to design your experiment. This is important because many of your inconsistencies in assembling a bulk clone tree stem from low-frequency mutations, but the low-VAF spectrum is where most of the neutral mutations reside; if those are removed how often is it that you remain with a non-identifiable treee?

Reviewer #3: The authors report on a new method, PhyDOSE, for determining the number of cells to sequence in a single-cell sequencing experiment based on information from bulk data. The bulk data is first used to estimate the mutation frequencies and then this information is used to estimate the number of cells. The authors state that their method improves upon SCOPIT, since the latter assumes knowledge about the number of clones and the frequency of the smallest clone. The authors study the performance of their method on simulated and empirical datasets.

I would like the authors to address the following questions:

1. How does the reliability of the mutations called from the bulk data affect the performance of the method? What if some/many of those mutations were wrong?

2. Why not compare to SCOPIT? After all, there are method for estimating clonality from bulk data. Why not run such a method, get the number of clones and frequencies, and use those as inputs to SCOPIT? I think it's very important to do this comparison.

3. I think the model of evolution must be incorporated into the problem formulation, as the number of cells and mutations needed depends on whether the infinite-sites assumptions holds or not.

My main issue with this method (which applies to SCOPIT as well, I feel, even though I don't know the details of how SCOPIT works) is that the number of cells to sequence is not the only/main quantity of interest in in an SCS experiment. The number of spatial regions to sample and sequence in order to capture the heterogeneity is as important, and that number must be a lower bound on the number of cells to sequence. So, I'm not sure how useful these methods will be in practice. Yes, SCOPIT has been published for a year only, but it still has no real citations (the two citations it has are this article and one that develops a simpler method for scRNA data). As scDNAseq becomes even less expensive, I doubt the number of cells is the bottleneck; it's the spatial regions to sample and sequence (indeed, some recent studies, mainly focused on CNA detection, are now sequencing thousands of single cells).

**Have all data underlying the figures and results presented in the manuscript been provided?**

Reviewer #1: Yes

Reviewer #2: Yes

Reviewer #3: Yes

PLOS authors have the option to publish the peer review history of their article (what does this mean?). If published, this will include your full peer review and any attached files.

Reviewer #1: No

Reviewer #2: No

Reviewer #3: No
---

## [Decision Letter · Decision Letter 1]

12 Aug 2020

Dear Dr. El-Kebir,

We are pleased to inform you that your manuscript 'PhyDOSE: Design of Follow-up Single-cell Sequencing Experiments of Tumors' has been provisionally accepted for publication in PLOS Computational Biology.

Best regards,

Niranjan Nagarajan

Associate Editor

PLOS Computational Biology

Florian Markowetz

Deputy Editor

PLOS Computational Biology

Reviewer's Responses to Questions

**Comments to the Authors:**

Reviewer #1: The revisions have satisfied all of my concerns about the paper. The authors have made a number of significant improvements to address the original criticisms, including extending the theory and code to multiple samples and uncertainty in the inferences, and empirically demonstrating robustness to data errors and violations of the infinite sites model, along with other more minor changes. The new material is substantial and very responsive to the critiques. I do not see any new problems introduced by the new material or have any other issues to raise. As before, I consider this an innovative and technically rigorous contribution to important current problems in cancer genomics that should be of interest to many working in computational biology or cancer research.

Reviewer #2: The authors presented an extended version with a new technical improvement to handle multiple biopsies of the same tumour. The formulation of the new problems follows from the single-sample ILP one. A new heuristic is proposed to solve some algorithmic complexity issues in some cases, but in general this is reasonable given the problem complexity.

This improvement approaches a point was shared also by another reviewer, and hasI feel it has been addressed properly.

I also asked to motivate practically the advantage of using this tool, showing the drop in costs in doing a proper experimental design with this new tool. The authors have motivated this for one case study that uses Tapestri; I never used Tapestri so I cannot confirm the reported costs, but the advantage is evident and I think this might be an important motivation to use this approach to designs Cancer Evolution assays that need to find the exact tumour clone tree.

Reviewer #3: The authors have done an excellent job responding to all comments, revising the writing, and running more experiments that I believe have strengthened the paper significantly.

I'm satisfied with it.

Very minor comments: What are A and B in panels (c) and (d) of Fig. 4? This should be described in the caption.

**Have all data underlying the figures and results presented in the manuscript been provided?**

Reviewer #1: Yes

Reviewer #2: None

Reviewer #3: Yes

PLOS authors have the option to publish the peer review history of their article (what does this mean?). If published, this will include your full peer review and any attached files.

Reviewer #1: No

Reviewer #2: No

Reviewer #3: No

---

## [Editor Report · Acceptance letter]

23 Sep 2020

PCOMPBIOL-D-20-00693R1 

PhyDOSE: Design of Follow-up Single-cell Sequencing Experiments of Tumors

Dear Dr El-Kebir,

I am pleased to inform you that your manuscript has been formally accepted for publication in PLOS Computational Biology. Your manuscript is now with our production department and you will be notified of the publication date in due course.

With kind regards,

Laura Mallard
